# PARP Inhibitors for Breast Cancer: Germline *BRCA1/2* and Beyond

**DOI:** 10.3390/cancers14174332

**Published:** 2022-09-05

**Authors:** Maria Clara Saad Menezes, Farah Raheem, Lida Mina, Brenda Ernst, Felipe Batalini

**Affiliations:** 1Harvard T.H. Chan School of Public Health, 677 Huntington Ave, Boston, MA 02115, USA; 2Mayo Clinic Arizona, 5777 E. Mayo Blvd, Phoenix, AZ 85054, USA

**Keywords:** breast cancer, PARPi, BRCA1, BRCA2, PALB2, homologous recombination repair, HRD

## Abstract

**Simple Summary:**

Poly-adenosine diphosphate ribose polymerase (PARP) inhibitors (PARPi) are effective against tumors with mutations in DNA repair genes, most commonly in the *BRCA1* and *BRCA2* genes. Because these tumors are unable to repair their DNA, PARPi have been used to target DNA repair pathways and are useful in the treatment of breast cancers with some of these alterations. There are two FDA-approved PARPi for patients with breast cancer—olaparib and talazoparib. The data on olaparib and talazoparib in the treatment of breast cancer are summarized in this review, and we also explore potential future applications of PARPi beyond inherited *BRCA* mutations.

**Abstract:**

Poly-adenosine diphosphate ribose polymerase (PARP) inhibitors (PARPi) are approved for *BRCA1/2* carriers with HER2-negative breast cancer in the adjuvant setting with a high risk of recurrence as well as the metastatic setting. However, the indications for PARPi are broader for patients with other cancer types (e.g., prostate and ovarian cancer), involving additional biomarkers (e.g., *ATM*, *PALB2*, and *CHEK*) and genomic instability scores. Herein, we summarize the data on PARPi and breast cancer and discuss their use beyond *BRCA* carriers.

## 1. Introduction

Poly-adenosine diphosphate ribose polymerase (PARP) inhibitors (PARPi) are effective against tumors with an impaired ability to repair double-strand DNA breaks, such as those with homologous recombination repair (HRR) deficiency (HRD) [1,2]. PARP enzymes play a role in a range of cellular activities. PARP1 and PARP2 are essential for the repair of single-strand breaks in DNA. When PARP enzymes are suppressed, DNA single-strand breaks accumulate and lead to DNA double-strand breaks at replication forks. When the processes for repairing double-strand breaks are inadequate, such as in tumor cells with HRD, there is a threat to cell survival. PARPi and HRD represent a lethal combination, which is the basis of the concept of synthetic lethality, though neither is lethal alone (Figure 1) [1,2,3]. In addition to the catalytic inhibition of PARP enzymes, PARPi trap PARP1 and PARP2 at damaged DNA sites, preventing the recruitment of additional DNA repair proteins and ultimately leading to cell death [4]. Trapped PARP–DNA complexes have been shown to be more cytotoxic than the single-strand breaks generated by PARP inactivation [4]. Differences in PARP trapping potential are not correlated with the inhibition of PARP catalytic activity [4].

PARPi have been extensively studied in cancers where HRD is prevalent, including breast cancer (BC). The development of BC is linked to germline and somatic pathogenic mutations in DNA-repair genes. Approximately 10% of BC cases are familial, and half of these are due to an inherited deleterious *BRCA1/2* mutation [5,6]. Familial BCs are overrepresented among women with triple-negative BC (TNBC): nearly 14% have pathogenic germline variants, and approximately 50% consist of *BRCA1/2* mutations [6]. While *BRCA1* and *BRCA2* pathogenic variants are associated with a high risk of BC, *PALB2* pathogenic variants are associated with a moderate risk of BC [7,8]. Variants in other DNA-repair genes are linked to a higher risk of estrogen-receptor-positive BC (*CHEK2*, *ATM*, and *CDH1*) as well as TNBC (*BARD1*, *RAD51C*, and *RAD51D*) [7,8]. The estimated odds ratios of BC linked with various DNA-repair mutations are shown in Table 1 [8].

For *BRCA1/2* carriers with human epidermal growth factor receptor two (HER2)-negative BC, PARPi can be beneficial in the adjuvant setting for patients at high risk of recurrence [10] and in the metastatic setting [11,12]. There are twelve other FDA-approved indications for PARPi in a variety of cancers, as summarized in Table 2 [11,12,13,14,15,16,17,18,19,20,21,22,23,24,25,26,27,28,29].

Currently, there are four FDA-approved PARPi: olaparib, rucaparib, niraparib, and talazoparib [14,15,16,17]. Olaparib and talazoparib are approved for BC patients [11,12]. In a phase II trial of rucaparib in *BRCA* carriers with advanced BC, there was no response among the 23 patients treated [30]. Olaparib and rucaparib have similar potencies for trapping PARP–DNA complexes [31], and all PARPi are pharmacologically similar in terms of inhibiting PARP catalytic activity. However, talazoparib is 100 times more effective at trapping PARP–DNA complexes and is more cytotoxic as a single agent than olaparib [31]. The varying initial doses of each PARPi drug reflect this relative variance in potency. The typical starting doses are 300 mg B.I.D. for olaparib; 200 or 300 mg daily for niraparib, depending on the patient’s baseline weight and/or platelet count; 600 mg B.I.D. for rucaparib; and 1 mg daily for talazoparib [14,15,16,17]. 

The present article reviews the key clinical trial data for the PARPi currently approved for BC—olaparib and talazoparib. We also discuss the role of PARPi in patient populations with BC harboring HRD mutations beyond *BRCA*.

## 2. Methods

We searched PubMed on 29 April 2022, for clinical studies exploring the use of PARPi in patients with BC using the following search terms: “breast” AND “Olaparib OR AZD2281 OR Talazoparib OR BMN 673”. The references of the included articles were also screened for eligible papers (Figure 2). Seventy-three trials were included and divided into the following groups: (Section 3) OLAPARIB, (Section 3.1) early-phase studies, (Section 3.2) locally advanced or metastatic BC, (Section 3.3) early-stage BC, and (Section 3.4) combination trials; (Section 4) TALAZOPARIB, (Section 4.1) early-phase studies, (Section 4.2) locally advanced or metastatic BC, (Section 4.3) early-stage BC.

## 3. Olaparib

### 3.1. Early-Phase Studies

Olaparib belongs to the N-acylpiperazine class and is made via the formal condensation of the free amino group of N-(cyclpropylcarbonyl)piperazine with the carboxy group of 2-fluoro-5-[(4-oxo-3,4-dihydrophthalazin-1-yl)methyl]benzoic acid [32]. It is a PARPi that targets PARP1, PARP2, and PARP3 [14]. Increased cytotoxicity and anti-tumor activity were observed in cell lines and mice tumor models with defects in *BRCA1/2*, *ATM*, or other genes involved in DNA repair after treatment with olaparib, and this was linked with platinum responsiveness [19,20]. In terms of monotherapy, phase I trials identified the maximum tolerated dose of olaparib capsules to be 400 mg B.I.D [33,34]. The mean maximal PARP inhibition in human peripheral blood mononuclear cells and tumor tissue is 50.6% and 70.0%, respectively [35]. Based on pharmacokinetics, tolerability, and efficacy measured by tumor shrinkage, the recommended olaparib monotherapy tablet dose is 300 mg B.I.D [36,37]. Olaparib was originally available in tablets and capsules. These dosage forms are not bioequivalent and thus are not interchangeable [37]. In the United States, capsule formulation was discontinued as of 2018. Olaparib absorption is delayed with high-fat meals, but the extent of absorption is not significantly altered [38,39]. Thus, olaparib can be administered with or without food, although administration with a meal may help prevent gastrointestinal adverse events (AEs) such as nausea or vomiting [38,39]. Olaparib is a major substrate of CYP3A4 and is primarily hepatically metabolized through oxidation [40]. Concomitant administration with strong inducers and inhibitors of CYP3A4 should be avoided [41]. If administration with a strong CYP3A4 inhibitor cannot be avoided, the olaparib tablet dose should be reduced to 100 mg B.I.D [14,41]. Olaparib is the only PARPi that does not cause transaminitis [42]. Olaparib has the most extensive hepatic metabolism and should be avoided when used with other agents that affect or undergo hepatic metabolism [40,43]. Pharmacokinetic studies showed that the mean area under the receiver operating characteristic curve (AUC) and peak serum concentration (C_max_) of olaparib were increased by 15% and 13%, respectively, in patients with mild hepatic impairment [43]. However, dose adjustment for mild or moderate hepatic impairment is not necessary [40]. Approximately, 44% of olaparib is excreted in the urine, mostly as metabolites. Exposure to olaparib was shown to be increased in renal impairment. In those with mild impairment (creatinine clearance (CrCl) 51 to 80 mL/min) AUC and C_max_ increased by 24% and 15%, respectively) [44]. Drug exposure increased to a higher extent (AUC and C_max_ increase by 44% and 26%, respectively) with moderate impairment (CrCl 31 to 50 mL/min), which required adjusting the olaparib dose (tablets) to 200 mg B.I.D [44]. An increase in serum creatinine was also observed with olaparib (up to 99%) [45,46]. It is believed that elevations in serum creatinine might not reflect a true decline in the glomerular filtration rate or kidney insufficiency, and monitoring alternative markers, such as Cystatin C, that are not impacted by transporters of creatinine should be considered to avoid an unnecessary dose reduction of olaparib and other PARPi [42,45,46]. Secondary hematological malignancy has rarely been reported in patients treated with olaparib [47]. The median duration of therapy prior to the development of the secondary cancers was two years (range: six months to >ten years) [14].

The most common AEs with olaparib include fatigue (67%), nausea (45% to 77%; grades 3/4: ≤3%), abdominal pain (34% to 45%), anemia (23% to 44%; grades 3/4: 7% to 21%), and neutropenia (5% to 19%; grades 3/4: 4% to 6%), with rashes (5 to 15%) and pneumonitis (<1%) being less common AEs [14]. Olaparib has no clinically significant effect on the QT interval [48]. Prolonged hematologic toxicity should prompt olaparib treatment interruption and the weekly monitoring of blood counts until recovery [14]. If counts do not recover to ≤ grade 1 after four weeks, further evaluation including bone marrow and cytogenetic analyses is necessary [14].

In combination therapy studies, olaparib has been studied with bevacizumab [49], cediranib [50,51,52], paclitaxel [53,54,55], carboplatin [56,57,58,59], a carboplatin/paclitaxel combination [60,61,62], cyclophosphamide [63], liposomal doxorubicin [64], cisplatin [65], lurbinectedin [66], durvalumab [51,52,67], dacarbazine [68], eribulin [69], ceralasertib [70], onalespib [71], prexasertib [72], gemcitabine [73], a cisplatin/gemcitabine combination [74], topotecan [75], and radiation therapy [76]. Olaparib was also studied in combination with phosphatidylinositol 3-kinase (PI3K) inhibitors (PI3Ki). While combination therapy required a dose attenuation of the pan-PI3Ki BKM120 in one study [77], the combination of olaparib with the PI3Ki alpelisib was shown to be safe and effective [78]. Similarly, olaparib combination appeared to be safe with the protein kinase B inhibitor capivasertib [79,80]. Additionally, there were no clinically relevant interactions between olaparib and endocrine therapy including anastrozole, letrozole, or tamoxifen [81]. 

### 3.2. Locally Advanced or Metastatic Breast Cancer

Phase II trials demonstrated the efficacy of PARPi in patients with BC. In patients that had a median of three prior treatments with chemotherapy, Tutt et al. demonstrated an objective response rate (ORR) of 41% and 22% in *BRCA* carriers who received olaparib at the doses of 400 mg B.I.D. and 100 mg B.I.D., respectively [82]. Kaufman et al. reported an ORR of 12.9% in 62 BC patients who received olaparib 400 mg B.I.D. with germline *BRCA* mutation (*gBRCA*) and three or more previous lines of therapy [21]. Conversely, Gelmon et al. investigated olaparib in advanced solid tumors, and no objective responses were reported in the 26 patients with BC (16 patients with TNBC and ten patients with *BRCA*-mutated BC). The median of previous therapies was also three [83].

In the phase III trial OlympiAD, patients with advanced HER2-negative BC and confirmed or suspected deleterious *gBRCAm* who had received no more than two previous chemotherapy regimens for metastatic disease and at least one endocrine therapy for hormone-receptor-positive disease were assigned to either olaparib monotherapy or standard-of-care chemotherapy. Patients were given olaparib tablets (300 mg B.I.D.) or standard therapy with a single-agent chemotherapy of their doctor’s choice in a 2:1 ratio (capecitabine, eribulin, or vinorelbine in 21-day cycles). Olaparib increased median progression-free survival (PFS) by nearly three months (7.0 months vs. 4.2 months; hazard ratio (HR) for disease progression or death, 0.58; 95% confidence interval (CI), 0.43–0.80; *p* < 0.001). With olaparib, the ORR was 59.9%, and with chemotherapy, it was 28.8%. The rate of grade 3 or higher AEs was 36.6% in the olaparib group and 50.5% in the conventional therapy group. Treatment discontinuation due to toxicity occurred in 4.9% and 7.7% of patients, respectively. There were no reports of myelodysplastic syndrome (MDS), acute myeloid leukemia (AML), or other secondary malignancies [12,84,85]. The FDA approved olaparib for the treatment of patients with *gBRCAm* and HER2-negative metastatic BC who have been treated with chemotherapy in the neoadjuvant, adjuvant, or metastatic setting, based on the findings of this study. Patients with hormone-receptor-positive disease should have received and progressed on a prior endocrine therapy [14]. The results of a recent phase IIIb trial were consistent with previous findings and reported an ORR of 50% in a similar patient population [86].

### 3.3. Early-Stage Breast Cancer

In the phase II PETREMAC trial, olaparib monotherapy demonstrated an ORR of 56.3% (18 out of 32) in patients who received olaparib for up to ten weeks before chemotherapy. Patients had previously untreated stage II/III TNBC with a tumor size of more than two cm (the median pretreatment tumor size was six cm) [87]. 

In the adjuvant setting, results from the OlympiA trial led to the FDA approval of olaparib for patients with *gBRCAm* in the curative intent setting. OlympiA was a randomized, double-blind, phase III trial that compared olaparib to a placebo in the adjuvant context for *BRCA* carriers with HER2-negative BC who had a high risk of recurrence. Patients who had received neoadjuvant chemotherapy were not allowed to receive additional chemotherapy following surgery. Patients with TNBC who received neoadjuvant chemotherapy had to have residual disease, while those who received adjuvant chemotherapy had to have positive axillary lymph node involvement or a primary tumor measuring at least two cm. Patients with a hormone-receptor-positive tumor who received adjuvant chemotherapy were required to have at least four positive lymph nodes, and those who received neoadjuvant chemotherapy were required to have not achieved a pathological complete response (pCR), with a clinical and pathological stage plus estrogen receptor status and histological grade (CPS + EG) score of at least three. The CPS + EG is a validated staging system for disease-specific survival that provides a prognosis assessment of patients with early-stage BC after treatment with neoadjuvant chemotherapy. The CPS + EG score estimates the probability of disease relapse based on pretreatment clinical stage and post-neoadjuvant CPS as well as the status of the estrogen receptor and histological grade. Scores range from zero to six, with higher scores indicating a worse prognosis [88].

At the three-year mark, in comparison to placebo, patients who received olaparib had higher rates of invasive disease-free survival (85.9% vs. 77.1%; HR, 0.58, 99.5% CI 0.41–0.82; *p* < 0.001) and distant disease-free survival (87.5% vs. 80.4%; HR, 0.57, 99.5% CI 0.39–0.83; *p* < 0.001) [10]. Grade 3 or higher AEs that occurred in the olaparib arm included anemia (8.7%), decreased neutrophil count (4.8%), decreased white-cell count (3.0%), fatigue (1.8%), and lymphopenia (1.2%). In the olaparib arm, 25% of patients had a dose reduction, compared with 5.2% in the placebo arm. The rate of MDS or AML was 0.2% (two patients) in olaparib arm versus 0.3% (three patients) in the placebo arm [10]. At a median follow up of 3.5 years, adjuvant olaparib significantly improved overall survival (OS), with a HR of 0.68 (98.5% CI 0.47–0.97; *p* < 0.01). The 4-year OS rate was 89.8% with olaparib vs. 86.4% with the placebo. Improvements in distant (HR 0.61; 95% CI 0.48–0.77) and invasive (HR 0.63; 95% CI 0.50–0.78) disease-free survival were sustained. There were no emergent AEs and no new cases of MDS or AML reported [89]. Based on these results, the treatment guidelines recommend one year of adjuvant olaparib for patients with HER2-negative, early-stage BC and *gBRCAm* who meet the criteria for high recurrence risk as defined in the OlympiA trial enrollment criteria discussed above [90].

### 3.4. Combination Trials

#### 3.4.1. Olaparib and Eribulin

Yonemori et al. published a phase I/II study in advanced or metastatic TNBC that considered the combination of olaparib (tablets) with eribulin. The recommended phase II dose for olaparib was determined to be 300 mg B.I.D. and 1.4 mg/m^2^ for eribulin. The median number of treatments given to the 24 patients in the phase II group was 5.5 (range: 1–28). Neutropenia (83.3%), leucopenia (83.3%), anemia (41.7%), febrile neutropenia (33.3%), and thrombosis were among the grade 3 AEs (8.3%). The response rate was 29.2% (7/24; 90% Cl 14.6–47.9). Despite the anticancer effect of the combination therapy, substantial rates of febrile neutropenia limited its use in clinical practice [69]. The median PFS and OS were 4.2 months (95% CI, 3.0–7.4) and 14.5 (95% CI, 4.8–22.0), respectively. Responders were enriched for tumors with homozygous *BRCA1*-promoter methylation, which may improve the accuracy of identifying TNBC patients who will benefit from the olaparib/eribulin combination therapy [91].

#### 3.4.2. Olaparib and Paclitaxel

In a randomized phase II trial, Fasching et al. compared the efficacy of paclitaxel and olaparib to paclitaxel/carboplatin followed by epirubicin/cyclophosphamide as a neoadjuvant treatment in patients with HER2-negative early BC and HRD (60.4% of patients were *BRCA* carriers, and the rest had a high HRD score based on the Myriad MyChoice HRD results). With paclitaxel/olaparib, the pCR rate was 55.1% (90% CI 44.5–65.3%), while with paclitaxel/carboplatin, it was 48.6% (90% CI 34.3–63.2%) [92].

#### 3.4.3. Olaparib and Durvalumab

A phase I/II trial of durvalumab with olaparib in solid tumors was published by Domchek et al. in 2020. Patients with *gBRCAm* and metastatic, HER2-negative, progressing BC were included in the study. The use of up to two previous lines of chemotherapy for metastatic BC was permitted. Olaparib (tablet) at 300 mg B.I.D. was administered for four weeks, followed by an intravenous infusion of olaparib 300 mg B.I.D. and durvalumab 1.5 g every four weeks until disease progression. Thirty-four patients were enrolled, and 11 (32%) of them had grade 3 or worse AEs, with anemia (four—12%), neutropenia (three—9%), and pancreatitis (two—6%) being the most common. Three patients (9%) stopped taking the medication due to side effects, while four patients (12%) had major side effects. Major side effects included dyspnea (one event), pancreatitis (two events), and immune-mediated events at the discretion of the investigator (one event). There were no deaths due to the treatment. Among the 30 patients eligible for efficacy analysis, the ORR at week 12 was 63.3% (90% CI 48.9–80.1%), and 24/30 (80%; 90% CI 64.3–90.9%) had disease control at 12 weeks [67].

#### 3.4.4. Olaparib, Durvalumab, and Paclitaxel

Pusztai et al. published the results of one arm of their phase II I-SPY2 adaptive platform study in 2021, which considered the combination of durvalumab and olaparib with weekly paclitaxel for the neoadjuvant treatment of stage II/III, HER2-negative BC. Weekly paclitaxel was administered with olaparib 100 mg B.I.D. on weeks one through eleven and intravenous durvalumab 1500 mg every four weeks. Weekly paclitaxel was followed by doxorubicin with cyclophosphamide in the control arm. The durvalumab/olaparib/paclitaxel arm contained 73 patients, while the standard-of-care paclitaxel control arm contained 299 patients. In all HER2-negative (20–37%), TNBC (27–47%), and hormone-receptor-positive/HER2-negative (14–28%) patients, durvalumab/olaparib/paclitaxel was linked to a higher pCR rate. In the durvalumab/olaparib/paclitaxel arm, 12.3% of patients had immune-related grade 3 AEs, compared to 1.3% in the control arm [55].

## 4. Talazoparib

### 4.1. Early-Phase Studies

Talazoparib is a heterocyclic compound with one ring that belongs to the class of phthalazines, with the molecular formula C_19_H_14_F_2_N_6_O [93]. Talazoparib is a potent PARPi, demonstrating both the strong catalytic inhibition of PARP1 and PARP2 and significant PARP trapping potential [31]. The maximum tolerated dose of talazoparib was determined to be one mg daily, but sustained PARP inhibition was reported at dosages as low as 0.60 mg/day [94,95]. Talazoparib is largely eliminated by the kidneys after limited hepatic metabolization [96,97]. Combination trials have included carboplatin, which caused significant hematologic toxicity [98]; temozolomide [99]; and irinotecan [100]. 

Talazoparib is only available in capsules, which can be administered with or without food [17]. Talazoparib is a major substrate of p glycoprotein/ABCB1, which is an ATP-dependent efflux pump [17]. Certain medications may inhibit or increase the serum concentration of p-glycoprotein, and screening for interactions is necessary [17]. Talazoparib undergoes minimal hepatic metabolism [96,97]. Renal excretion accounts for 69% of the drug clearance, and up to 54.6% of the drug is excreted unchanged in the urine [96,97]. Talazoparib exposure is increased by 12%, 43%, and 163%, and C_max_ is increased by 11%, 32%, and 89% with mild (eGFR 60 to 89 mL/minute/1.73 m^2^), moderate (eGFR 30 to 59 mL/minute/1.73 m^2^), and severe (eGFR 15 to 29 mL/minute/1.73 m^2^) renal impairment, respectively [96,97]. As a result, talazoparib requires a renal dose adjustment to 0.75 mg daily for moderate renal impairment (CrCl 30 to 59 mL/min) and 0.5 mg daily for severe renal impairment (CrCl 15 to 29 mL/min) [96,97]. Common AEs include decreased hemoglobin (90%; grade 3: 39%), anemia (53%; grade 3: 38%), neutropenia (35%; grade 3: 18%), thrombocytopenia (27%; grade 3: 11%), fatigue (62%), nausea (49%), headache (33%), and transaminitis (37%) [17]. A higher talazoparib concentration is linked to an increased risk of anemia and thrombocytopenia [101]. Alopecia was reported in 25% of patients treated with talazoparib [17]; however, this was classed as grade 1 (<50% hair loss that is not obvious from a distance) and was considered hair thinning. Grade 2 alopecia (>50% hair loss that is apparent from a distance) was reported in only 2.4% of patients [102,103]. Serum creatinine elevation has not been reported with talazoparib [17]. Talazoparib has less emetogenic potential (minimal to low) than olaparib, niraparib, and rucaparib, which are associated with a moderate to high emetic risk [104]. The reported rate of MDS/AML with talazoparib is less than one percent [17]. The duration of talazoparib therapy prior to the development of MDS/AML ranges from four months to five years [17].

### 4.2. Locally Advanced or Metastatic Breast Cancer

The ABRAZO trial examined whether talazoparib could help *BRCA* carriers with locally advanced or metastatic cancer who had previously been exposed to platinum treatment. Those who had previously received platinum-based chemotherapy had an ORR of 21%, while patients who had received at least three non-platinum-based regimens had an ORR of 37% [105,106]. Litton et al. published the results of the EMBRACA trial in 2018, which was a randomized, open-label, phase III trial with advanced *gBRCAm* BC patients who had received three or fewer cycles of chemotherapy for metastatic disease. Two hundred and eighty-seven patients were given talazoparib (one mg per day) and 144 patients were given standard single-agent chemotherapy (capecitabine, eribulin, gemcitabine, or vinorelbine). Patients who had previously undergone platinum treatment were eligible if they had a disease-free interval of at least six months and no signs of progression on previous platinum therapy. The talazoparib group had a significantly longer PFS than the conventional therapy group (8.6 months vs. 5.6 months; HR 0.54; 95% CI, 0.41–0.71; *p* < 0.001) [11]. Based on the results of this study, the FDA approved talazoparib for the treatment of patients with *gBRCAm* and HER2-negative locally advanced or metastatic BC [17]. 

### 4.3. Early-Stage Breast Cancer

In a pilot study of talazoparib for early-stage BC, following two months of talazoparib preoperative monotherapy, all thirteen *BRCA* carriers included showed a decreased tumor volume. The average decrease in volume was 78% (range 30–98%). Talazoparib was well-tolerated, with no grade 4 side effects reported, and only one patient’s dose had to be reduced due to grade 3 neutropenia [107]. Litton et al. reported in 2020 that neoadjuvant talazoparib treatment for six months resulted in a pCR of 53% in patients with *gBRCAm* and operable stage I to III BC [108].

## 5. Biomarkers of Response to PARPi

To date, conflicting results for different proposed HRD biomarkers based on copy-number variations have been published. Telli et al. reported that a combined HRD score—unweighted sum of loss of heterozygosity (LOH), telomeric–allelic imbalance, and large-scale state transition scores—predicted responses in three neoadjuvant TNBC trials of platinum-containing therapy [109]. In contrast, TBCRC-030 [110] showed that the Myriad MyChoice HRD results were not predictive of pathologic responses to platinum agents. Additionally, Blum et al. [111] showed that *BRCA* LOH status, DNA damage response and repair gene mutational burden, and genome-wide LOH were not associated with responses to talazoparib in *BRCA* carriers enrolled in the EMBRACA trial. 

Batalini et al. showed that mutational signature 3 (Sig3) and the genomic instability score (GIS) were associated with responses to olaparib and that Sig3 demonstrated overall better performance than GIS for identifying responders [112]. The Spanish NOBROLA trial is currently enrolling non-*BRCA* patients that have a high genome-wide LOH (ClinicalTrials.gov identifier: NCT03367689). The PETREMAC trials evaluated several potential biomarkers of response to PARPi: *BRCAness*, *PAM50* gene expression, *RAD51* foci, tumor-infiltrating lymphocytes, and programmed cell death ligand one analyses were performed on pretreatment samples. Somatic or germline mutations affecting HRR pathway genes were observed in 10/18 responders (55.6%, 95% CI 33.7–75.4), in contrast to the 1/14 in non-responders. Among tumors without HRR pathway mutations, 6/8 responders vs. 3/13 non-responders revealed *BRCA1* hypermethylation (*p* < 0.04). Thus, 16/18 responders (88.9%, 95% CI 67.2–96.9), in contrast to 4/14 non-responders (28.6%, 95% CI 11.7–54.7; *p* < 0.001), carried HRR pathway mutations and/or *BRCA1* methylation [87]. 

In 2021, Patsouris et al. published a phase II trial that included 42 patients, 40 of whom received at least one dose of rucaparib [113]. The study assessed the efficacy of rucaparib in HER2-negative metastatic BC with either a high genome-wide LOH score or somatic *BRCA* mutation [113]. The study was powered to detect a 20% clinical benefit rate [113]. The primary endpoint was not reached, with a clinical benefit rate of 13.5% [113]. Two high-LOH patients, without somatic *BRCA* mutation, presented a complete and durable response (the duration of response was 12 and 28.5 months in each patient) [113]. Whole-genome analysis was performed on 24 samples, including five patients who presented a clinical benefit from rucaparib [113]. HRDetect was associated with response to rucaparib, although without reaching statistical significance (median HRDetect responders vs. non-responders: 0.465 vs. 0.040, *p* > 0.2) [113]. Another phase II trial investigating rucaparib in untreated TNBC demonstrated that only 12% of patients had decreased Ki67 with treatment (primary endpoint) [114]. In secondary endpoint analyses, HRD was identified in 69% of TNBC patients with the mutational-signature-based HRDetect assay and confirmed by impaired *RAD51* foci formation [114]. Following rucaparib treatment, there was no association between HRDetect and a Ki67 change, but circulating tumor DNA was more suppressed in patients with the HRDetect signature [114]. This data suggest that a small subset of patients with high genome-wide LOH scores and without *gBRCAm* could derive benefit from PARPi [113,114].

## 6. Discussion

PARPi have revolutionized the therapeutic landscape of *BRCA*-related BC. Many PARPi are still being investigated. In addition to the FDA-approved drugs olaparib and talazoparib, niraparib has shown some activity in *BRCA* carriers with BC on the BRAVO study and is being studied in different combinations [115]. However, the efficacy of olaparib and talazoparib have not been replicated by other contenders. Phase I clinical trials have demonstrated olaparib and talazoparib to be well-tolerated with a relatively similar side-effect profile. Olaparib is administered twice daily, while talazoparib is administered once daily. Anemia of any grade is one of the most common AEs reported with both olaparib and talazoparib. While olaparib is more commonly associated with nausea, vomiting, and fatigue, talazoparib has higher rates of cytopenia. For *BRCA* carriers with advanced HER2-negative BC, phase III clinical trials have demonstrated improved tolerability and superior efficacy of PARPi monotherapy in comparison to standard chemotherapy [11,12]. For *BRCA* carriers with early HER2-negative BC at high risk for recurrence, adjuvant olaparib was shown to significantly improve the invasive disease-free survival and OS [10,89,116]. Clinical benefits were demonstrated in all *BRCA* carrier patients with TNBC, those with hormone-receptor-positive disease, and patients who received previous platinum-based chemotherapy.

Selecting an adjuvant regimen for *BRCA* carriers with TNBC who do not achieve pCR and are therefore at high risk for relapse, remains a current clinical challenge. The results of three clinical trials (i.e., CREATE-X, OlympiA, and KEYNOTE-522) need to be considered [10,117,118]. The CREATE-X trial assigned HER2-negative BC patients with residual illness after neoadjuvant therapy (roughly one-third of whom had TNBC) to either eight cycles of adjuvant capecitabine or no additional chemotherapy. The five-year disease-free survival (74% vs. 68%; HR 0.70, 95% CI 0.53–0.92; *p* < 0.02) and OS (89% vs. 84%; HR for death 0.59, 95% CI 0.39–0.90; *p* < 0.02) were greater in capecitabine patients. Capecitabine’s improvement in disease-free survival was attributed to better outcomes among TNBC patients, according to subgroup analyses (70% vs. 56%; HR 0.58, 95% CI 0.39–0.87) [118]. Patients with stage II or III TNBC receiving neoadjuvant therapy were randomized to receive pembrolizumab or placebo every three weeks during neoadjuvant chemotherapy and for another nine cycles (27 weeks) after surgery in the KEYNOTE-522 trial. Pembrolizumab increased the pCR rate from 51% to 65%, regardless of the presence or absence of PD-L1 expression [117]. Finally, the OlympiA trial found that adjuvant olaparib improved three-year disease-free survival from 77% to 86% in *BRCA* carriers with early-stage, high-risk HER2-negative BC compared to placebo, and led to a significant improvement in OS (HR, 0.68; 98.5% CI 0.47–0.97; *p* < 0.01) [10].

The clinical benefits of sequencing or combining olaparib with other treatments such as pembrolizumab or capecitabine for the adjuvant therapy of patients with TNBC and *gBRCAm* is uncertain. Olaparib has been evaluated in combination with the checkpoint inhibitor durvalumab, and concurrent therapy was found to be safe and effective. However, safety data on the use of olaparib concurrently with other drugs such as temozolomide, pembrolizumab, and capecitabine are limited. Ongoing clinical trials will evaluate the combination therapy of olaparib/pembrolizumab (NCT04191135, NCT05174832, NCT03025035, NCT05203445, NCT04683679, and NCT05033756) and olaparib/temozolomide (NCT05128734) in patients with BC. The combination of olaparib/capecitabine is yet to be investigated. 

Conversely, other PARPi were studied in combination with either pembrolizumab or temozolomide. Niraparib and pembrolizumab were examined in 55 patients with advanced TNBC, regardless of *BRCAm* status or PD-L1 expression, by Vinayak et al. An objective response was observed in ten patients (18% (five complete responses and five partial responses)). Anemia (ten—18%), thrombocytopenia (eight—15%), and fatigue (four— 7%) were the most common treatment-related AEs of grade 3 or above. Immune-related AEs were reported in eight individuals (15%), two of whom had grade 3 AEs (4%). There were no new safety signals identified [119]. For temozolomide, two studies had shown that the combination of a veliparib with temozolomide is safe without excessive toxicity [120,121]. Due to the varied potencies of different PARPi and varying myelosuppressive potential, safety results on veliparib combined with chemotherapy cannot be generalized to other PARPi.

A few clinical trials have now demonstrated the potential efficacy of PARPi beyond *BRCA* carriers. While the study by Gelmon et al. did not show responses in heavily pretreated patients in the metastatic setting [83], Eikesdal et al. showed a high response rate in treatment-naïve and unselected TNBC patients to olaparib monotherapy in the neoadjuvant setting [87]. In the phase II trial TBCRC-048, Tung et al. demonstrated high rates of response in patients with somatic *BRCA* mutation (ORR 50%) and germline *PALB2* mutations (ORR 82%) [122]. Confirmatory expansion cohorts of this study are currently enrolling patients, and results are awaited. Future analyses of data from TBCRC-048 may shed light on the biomarkers of PARPi response [122]. No responses were observed with *ATM* or *CHEK2* mutations alone [122]. Additionally, Batalini et al. showed that alpelisib and olaparib can lead to meaningful responses in heavily pre-treated patient populations [78], hence confirming the results of preclinical data that PI3Ki can render tumors sensitive to PARPi [123].

During treatment with PARPi, including olaparib and talazoparib, the complete blood count should be monitored with differential at baseline and monthly thereafter, or as clinically indicated with increasing frequency to weekly until recovery for prolonged hematologic toxicity [14,17]. Talazoparib and olaparib package inserts also require the monitoring of renal function without specifying the frequency of monitoring [14,17]. Since the dose modification of olaparib and talazoparib for renal function is required, the baseline assessment of renal function is necessary. There are no specific requirements for monitoring complete metabolic panel for either PARPi agents; however, a baseline assessment should be considered at minimum with an increase in monitoring during therapy if indicated, especially with PARPi agents that are associated with transaminitis. 

As a biomarker, Sig3 is advantageous to other proposed HRD biomarkers because it leverages clinical sequencing that is already routine—gene panel sequencing—without the need for an additional assay or sequencing technology [112,124]. Sig3 captures the different mechanisms associated with underlying HRD in BC, including the biallelic inactivation of *BRCA1*/2, germline nonsense and frameshift variants in *PALB2*, missense *BRCA1*/2 variants known to impair HRR pathway, and the epigenetic silencing of *RAD51C* and *BRCA1* by promoter methylation [125]. Accordingly, Batalini et al. demonstrated that Sig3 had an overall better performance than GIS for identify olaparib responders among BC patients [112]. However, prospective data are necessary to validate Sig3 as a useful biomarker.

## 7. Conclusions

PARPi monotherapy—olaparib and talazoparib—are approved for *BRCA* carriers with advanced or metastatic HER2-negative BC. Olaparib is approved in the adjuvant setting for *BRCA* carriers at a high risk of relapse [10]. While *BRCA* carriers constitute a minority of patients with BC, there is mounting evidence that PARPi could also benefit more patients [78,87,122]. The identification of a biomarker of response to PARPi remains a critical goal.

## Figures and Tables

**Figure 1 cancers-14-04332-f001:**
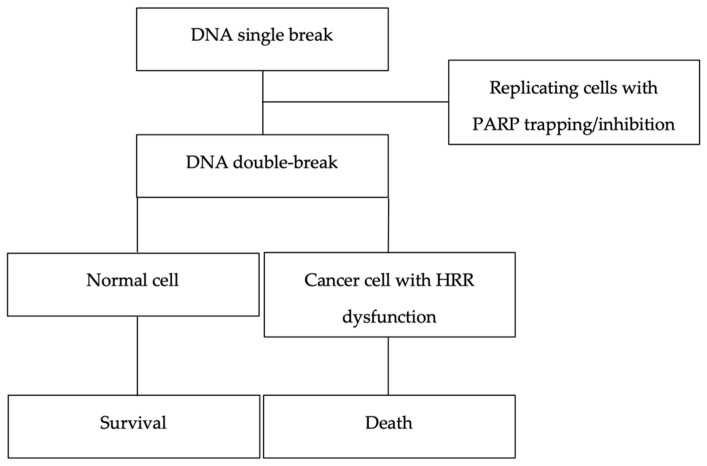
A synthetic lethality therapeutic approach: poly(ADP) ribose polymerase inhibitors (PARPi) for the treatment of cancers with a deficient homologous recombination repair (HRR) pathway. Neither PARPi nor HRR deficiency (HRD) alone is lethal, but the inadequate repair of double-strand breaks found in HRR-deficient cells renders them sensitive to PARP inhibition.

**Figure 2 cancers-14-04332-f002:**
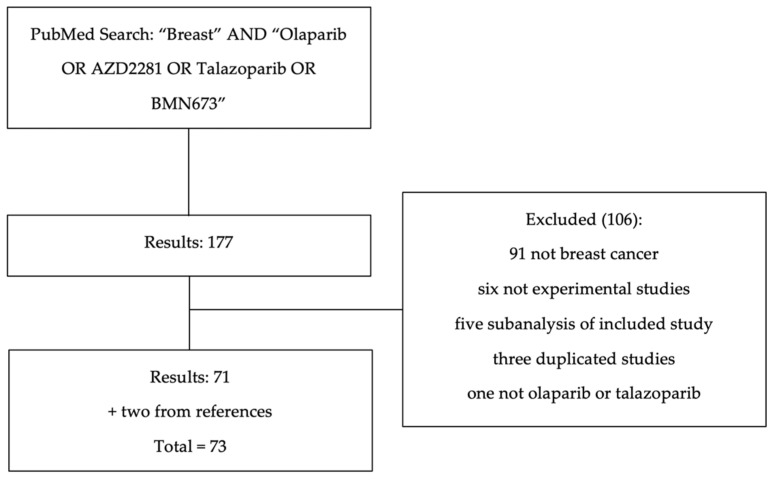
Search strategy. On 29 April 2022, the following PubMed search was performed: “breast” AND “Olaparib OR AZD2281 OR Talazoparib OR BMN673”. The references were also scanned for eligible studies. There were 177 results. Among them, 106 studies were excluded, as 91 cited the word breast in the body of the text but were not about breast cancer, 6 were not experimental studies, five were subanalyses of included studies, 3 were duplicated studies, and 1 was an experiment that included neither olaparib nor talazoparib. Accordingly, 71 studies were included, with two additional studies from the references.

**Table 1 cancers-14-04332-t001:** Associations between pathogenic variants in established breast-cancer-predisposition genes and risk of breast cancer.

Pathogenic Variant	Odds Ratio (95% CI)	*p*-Value
*BRCA1*	7.62 (5.33–11.27)	<0.001
*BRCA2*	5.23 (4.09–6.77)	<0.001
*PALB2*	3.83 (2.68–5.63)	<0.001
*ATM*	1.82 (1.46–2.27)	<0.001
*CHEK2*	2.47 (2.02–3.05)	<0.001

Adapted from reference [8], with loss-of-function variants and variants identified as “pathogenic” or “likely pathogenic” in the ClinVar [9] database.

**Table 2 cancers-14-04332-t002:** FDA indications for poly-adenosine diphosphate ribose polymerase inhibitors (PARPi) in a variety of cancers.

Drug	Indications [14,15,16,17]	Biomarker	Main Trial
**Olaparib**	Advanced epithelial ovarian *	*BRCA1*/2	SOLO-1 [13] (2018)
	Advanced epithelial ovarian *	*BRCA1*/2, or GIS	PAOLA-1 [18] (2020)
	Recurrent epithelial ovarian *	X	SOLO-2 [19] (2017), Study 19 [20] (2017)
	Advanced ovarian	*gBRCA1*/2	NCT01078662 [21] (2014)
	**Metastatic breast: HER2-negative**	***gBRCA1*/2**	**OlympiAD [12] (2018)**
	Metastatic pancreatic adenocarcinoma	*gBRCA1*/2	POLO [22] (2019)
	Metastatic prostate	*ATM*, *BRCA1*/2, *BARD1*, *BRIP1*, *CDK12*, *CHEK1*/2, *FANCL*, *PALB2*, *RAD51*, *RAD54L*	PROfound [23] (2020)
**Rucaparib**	Recurrent epithelial ovarian *	X	ARIEL3 [24] (2018)
	Epithelial ovarian *	*BRCA1*/2	Study 10 and ARIEL2 [25] (2016)
	Metastatic prostate	*BRCA1*/2	TRITON2 [26] (2020)
**Niraparib**	Advanced epithelial ovarian *	X	PRIMA [27] (2020)
	Recurrent epithelial ovarian *	X	NOVA [28] (2017)
	Advanced ovarian *	*BRCA1*/2 or GIS	QUADRA [29] (2019)
**Talazoparib**	**Metastatic/advanced breast: HER2-negative**	***gBRCA1*/2**	**EMBRACA [11] (2018)**

* Additionally, fallopian tube or primary peritoneal; *gBRCAm*: germline *BRCA* mutation; HER2: human epidermal growth factor receptor 2; GIS: genomic instability score. Indications for use in breast cancer patients are bolded.

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
