# Peer review of "PARP Inhibitors for Breast Cancer: Germline BRCA1/2 and Beyond"

_cancers, 2022, doi:10.3390/cancers14174332_

Round 1

Reviewer 1 Report

The authors have done decent work with summarizing the applications of PARP inhibitors in BRCA and non-BRCA settings. The authors have performed an extensive literature search and have summarized the critical information. I think the current draft is in a good shape however I do want the below points to be addressed.

·      Authors should incorporate more information/statistics on % of having germline BRCA gene mutation out of total breast cancer cases. 

·      I guess authors should provide the structure of different PARP inhibitors that are in clinical use. Also, there should be a separate section on what we know about the trapping efficiency of these small molecule inhibitors. There is adequate information available on PARPi has variable trapping efficiency vs cellular toxicity. 

·      Figures 1 and 2 require editing as the schematics are not uniformly made (see the size and position of arrows).

·      The figure legends (especially for fig 2) are very short. I think the figure legends should be descriptive that provide readers with enough information to comprehend the data. 

·      Lastly, I noticed the authors have missed citing a reference in many places for instance line 128-129 on page 5. In addition to this, authors should also pay attention to citing the actual classic article rather than citing a review.

Reviewer 2 Report

Please see the attached review report. Thanks.

Reviewer 3 Report

In this comprehensive review, the authors discussed the role of  PARP inhibitors (PARPi) in the treatment of BRCA-related breast cancer and introduced clinical trials in which FDA-approved PARPi drugs were involved. This paper will be a great reference for physician-scientists as well as clinicians in the breast cancer research and treatment field. I only have minor comments as followings.

1. In Table 1, the genetic variants corresponding to the odds ratios should also be presented. 

2. The title of Table 2 is exactly the same as that of Table 1, which should be wrong.

3. Some basic scientific writing errors, i.e. numbers less than ten should be spelled out,  'figure' and 'table' should be capitalized whenever cited in the main text, and inconsistent writing style in figures (some words fully capitalized while some not). 

4. Some abbreviations may not be as concise as possible. For example, AUC is short for area under the receiver operating characteristic curve, not simply 'area under the curve'. 

5. Some P values are exact values while others are ranges. Please be as consistent as possible.

Round 2

Reviewer 2 Report

I appreciate the authors’ efforts to revise and improve this paper. This revision looks good and acceptable to me.